# A Head-to-Head Comparison Between [^18^F]Fluorodeoxyglucose ([^18^F]FDG) Positron Emission Tomography/Computed Tomography (PET/CT) and ^99m^Technetium-Hexamethylpropylene Amine Oxime (HMPAO)-Labeled Leukocyte Scintigraphy in a Case Series of Patients with Suspected Vascular Prosthesis Infection: To Trust Is Good, but to Check Is Better

**DOI:** 10.3390/jcm14124352

**Published:** 2025-06-18

**Authors:** Marina Scarpuzza, Alice Ambrogio, Andrea Leo, Lorenzo Roberto Suardi, Michele Marconi, Marco Falcone, Raffaella Berchiolli, Elena Lazzeri

**Affiliations:** 1Regional Center of Nuclear Medicine, Medical School, University of Pisa, University Hospital of Pisa, Via Roma 67, 56126 Pisa, Italy; ambrogioalice@hotmail.it (A.A.); andrea.leo.due@gmail.com (A.L.); e_lazzeri@hotmail.com (E.L.); 2Infectious Diseases Unit, Department of Clinical and Experimental Medicine, University Hospital of Pisa, 56100 Pisa, Italy; lorenzoroberto.suardi@gmail.com (L.R.S.); marco.falcone@unipi.it (M.F.); 3Division of Vascular Surgery, University Hospital of Pisa, Via Paradisa 2, 56124 Pisa, Italy; michelemarconi@gmail.com (M.M.); raffaella.berchiolli@unipi.it (R.B.)

**Keywords:** prosthetic vascular infection, PVGI, [^18^F]FDG PET/CT, WBC scintigraphy, infection, MAGIC criteria

## Abstract

**Background:** Prosthetic vascular graft infection (PVGI) is a serious complication associated with vascular prostheses. Nuclear medicine techniques, including [^18^F]fluorodeoxyglucose positron emission tomography/computed tomography (PET/CT) and ^99m^technetium-hexamethylpropylene amine oxime (HMPAO)-labeled leukocyte (WBC) scintigraphy, are part of the MAGIC diagnostic criteria for PVGI. **Methods:** In this retrospective study, we analyzed eight patients with suspected PVGI who underwent both [^18^F]FDG PET/CT and WBC scintigraphy within an average of 8 days. **Results:** Of all eight patients (median age 69 years), three showed concordant positive results with both PET/CT and WBC, and their final diagnosis confirmed the presence of infection; five showed discordant results: in all five of these patients, PET/CT showed false-positive findings, whereas WBC correctly identified five true-negative cases. **Conclusions:** [^18^F]FDG PET/CT is highly sensitive but prone to false positives. WBC scintigraphy, combined with SPECT/CT, particularly in the evaluation of the treatment response, showed greater specificity, and it may warrant consideration as a MAGIC major diagnostic criterion for PVGI.

## 1. Introduction

Vascular graft infection is a devastating complication of vascular reconstructive surgery and represents one of the most frequent causes of morbidity and mortality in patients with vascular prostheses. The incidence of prosthetic vascular graft infection (PVGI) depends mainly on the anatomic location of the graft and on the implant procedures [1]. PVGI can be classified as early (within 4 months), largely due to intraoperative contamination, or late (after 4 months), mainly caused by the hematogenous spread of microorganisms, according to the onset after surgery. Accurate assessment of the presence of infection and of its causative agent is mandatory for an adequate therapeutic strategy in patients with suspected PVGI [1]. CT angiography (CTA) is the imaging of choice in patients with suspected PVGI, as it may show unequivocal signs of infection, such as the presence of fluid collection, gas, and soft tissue enhancement. However, when using CTA, it might not be possible to distinguish between infection and inflammation, especially in the case of low-grade infections [2,3,4]. Nuclear medicine procedures can improve the diagnostic accuracy of radiology-based imaging in PVGI. PET/CT with the glucose analog [18F]fluorodeoxyglucose ([^18^F]FDG) and scintigraphy with ^99m^Technetium-hexamethylpropylene amine oxime (HMPAO)-labeled leukocytes (WBCs) are the current nuclear medicine imaging techniques employed for infection diagnosis. [^18^F]FDG PET/CT has a high negative predictive value, but it should be performed at least 4 months after surgery to avoid false positive findings due to persisting inflammation, whereas WBC scintigraphy can be performed with satisfactory diagnostic accuracy at any time after surgery [5]. The combination of nuclear medicine diagnostic procedures with radiologic imaging has resulted in significantly increased sensitivity and specificity for the diagnosis of PVGI [6,7]. The Management of Aortic Graft Infection Collaboration (MAGIC) criteria (a combination of clinical/surgical, radiologic, and laboratory criteria) are often used for the PVGI diagnosis; their major criteria include CT-based findings, while minor criteria include [^18^F]FDG PET/CT and WBC scintigraphy [8]. A comparison of [^18^F]FDG PET/CT with WBC scintigraphy has previously been reported [9], based on a retrospective analysis of a cohort of 39 patients. The authors concluded that the diagnostic accuracy of WBC scintigraphy was greater than that of [^18^F]FDG PET/CT, especially regarding specificity; these results are somewhat discordant with a previous study [10] carried out in a much smaller cohort (11 patients), reporting no definite diagnostic advantage of one method over the other. Despite the results reported by Puges et al. on the better diagnostic accuracy of WBC scintigraphy versus [^18^F]FDG PET/CT [9], most of the studies published over the last decade are based on the use of [^18^F]FDG PET/CT as the radionuclide imaging procedure [7,11], perhaps because of the higher complexity of WBC scintigraphy in a busy clinical routine. In this scenario, our report aims to contribute to solving the diagnostic dilemma faced by the nuclear medicine specialists when a patient with suspected PVGI is referred for evaluation. Although based on a small group of patients that prevents statistical analysis, the cases presented here are meant to represent a variety of clinical conditions and imaging findings that can help other colleagues to develop their own approach in this kind of patient. This might help to identify the most appropriate use of either [^18^F]FDG PET/CT or WBC scintigraphy in the diagnostic work-up of such patients.

## 2. Materials and Methods

### 2.1. Patient Population

We retrospectively studied 8 adult male patients (mean age 67 years, median age 69 years, range 57–77 years) with suspected and diagnosed prosthetic vascular graft infection (PVGI) according to MAGIC criteria, evaluated between July 2022 and December 2023. All the clinical features of the patients were collected before performing [^18^F]FDG PET/CT and technetium-99 m HMPAO-labeled leukocyte (WBC) scintigraphy. The final diagnosis or exclusion of PVGI was based on a combination of diagnostic tests and/or clinical follow-up for at least six months. Written informed consent was obtained from all participants included in this study. This retrospective study was conducted in accordance with the Declaration of Helsinki and was approved by our Institutional Review Board (protocol code 9989, date of approval 20 February 2019). The analysis in our study is based on routinely performed imaging modalities; the local Institutional Review Board waived the need for approval.

### 2.2. Exclusion Criteria

Children, patients who did not undergo both nuclear imaging tests, and cases in which there was clinical, therapeutic, and/or surgical modification between the two examinations were not included. Although more than twenty patients with suspected prosthetic vascular graft infection (PVGI) were evaluated during the study period (July 2022–December 2023), the majority were excluded due to non-compliance with one or more of the above criteria. This rigorous selection was essential to ensure the reliability and comparability of the imaging findings.

### 2.3. Imaging Equipment and Radiopharmaceuticals [^18^F]FDG PET/CT

All patients underwent [^18^F]FDG PET/CT with a Discovery 710 scanner (GE Medical Systems, Waukesha, WI, USA). The administered activity of [^18^F]FDG (Gluscan^®^, Advanced Accelerator Applications Molecular Imaging Italy) was approximately 3.7 MBq/kg body weight. All patients were fasted for at least six hours, and their blood glucose values, assessed before [^18^F]FDG injection, were lower than 160 mg/dL. PET and CT images were acquired consecutively 60–90 min after i.v. tracer injection. CT data were used for both attenuation correction of PET emission data and fusion with attenuation-corrected PET images. An emission PET scan was obtained with 2-min acquisitions per bed position using a 3D acquisition mode. 

### 2.4. Radiolabeling of Autologous Leukocytes and WBC Scintigraphy 

All patients underwent WBC scintigraphy with a Discovery NM/CT 670 scanner (GE Healthcare, Waukesha, WI, USA). Autologous leukocytes were radiolabeled with ^99m^Tc-HMPAO (Exametascan, Radiopharmacy Laboratory Ltd., Budaörs, Hungary) according to the European Association of Nuclear Medicine (EANM) guidelines [12]. Radiolabeling efficiency ranged between 63% and 74%. The viability of radiolabeled leukocytes was routinely checked using the trypan blue exclusion test and was always >99%. ^99m^Tc-HMPAO WBC imaging was acquired with total-body and planar images of the region of interest 30 min (early), 4–6 h (delayed), and 20–22 h (late) after ^99m^Tc-HMPAO WBC i.v. injection (370–555 MBq). Additionally, SPECT/CT images of the region of interest were acquired at delayed and/or late times. Both CT attenuation-corrected and non-corrected (AC and NAC, respectively) images were analyzed. An Xeleris workstation (GE Healthcare, Oslo, Norway) was used to fuse the matching pairs of X-ray transmission and radionuclide emission images and to generate the hybrid images of overlying transmission and emission data. Images were displayed and analyzed in axial, coronal, and sagittal planes. MIP images were also available for review.

### 2.5. Interpretation Criteria

[^18^F]FDG PET/CT and WBC scintigraphy were assessed by two expert nuclear medicine physicians (A.A. and E.L.). [^18^F]FDG PET/CT was classified as negative when no uptake of [^18^F]FDG was observed in vascular and/or in perivascular prostheses, or when there was diffuse mild tracer uptake along the entire vascular graft. [^18^F]FDG PET/CT was classified as positive when the focal uptake of [^18^F]FDG in vascular and/or perivascular prosthesis tissues was present [13,14]. WBC scintigraphy was classified as negative for infection when no WBC accumulation was detected in vascular and/or at perivascular prosthesis tissues. Conversely, scintigraphy was classified as positive when one or more focal accumulations of WBC, increasing in intensity and/or changing in shape from delayed to late images, were observed in vascular and/or perivascular prosthesis tissues [5]. On the other hand, WBC scintigraphy was classified as negative for infection but consistent with inflammation when one or more focal accumulations of WBC, significantly decreasing over time (>20%), were present in vascular and/or perivascular prosthesis tissues [5]. The SPECT/CT coronal, transaxial, and sagittal planes, as well as MIP cine mode and both CT attenuation-corrected and non-corrected SPECT images, were evaluated. SPECT/CT imaging evaluation is essential in all patients with intracavitary PVGI, as the SPECT/CT images allow one to locate the site of infection, which can be masked by physiological bone marrow or accumulation of activity in the bowel.

### 2.6. Data Analysis

[^18^F]FDG-PET/CT and WBC scintigraphy results were compared directly. No comparative analysis was performed between the planar and SPECT/CT images or between the SPECT/CT and SPECT alone findings.

## 3. Results

We retrospectively analyzed the data obtained for eight patients (all men, mean age 67 years, median 69 years, range 57–77 years) with suspected PVGI, evaluated between July 2022 and December 2023. The main clinical features of the patients are summarized in Table 1. The final diagnosis or exclusion of PVGI was based on a combination of diagnostic tests and/or clinical follow-up for at least six months. Written informed consent was obtained from all participants included in this study. Our eight patients with suspected PVGI are described individually hereinafter.

Patient no. 1 (69 yrs.) had a Dacron prosthetic graft (Gelweave Dacron Graft, Terumo Aortic, Vascutek Ltd., Newmains Avenue, Inchinnan, Renfrewshire, PA4 9RR, UK) in the ascending aorta. The onset of fever of unknown origin (FUO) occurred twelve years after surgery [15]. Blood and urine cultures and transesophageal echocardiography were negative. Empirical antibiotic therapy (piperacillin/tazobactam and daptomycin) was administered for one week, ensuring complete remission of fever. Three months later, the patient was referred for [^18^F]FDG PET/CT because of the recurrence of fever with positive inflammation markers (ESR and CRP). The PET/CT scan showed significant focal increased [^18^F]FDG uptake (SUVmax 5.2) in the anterior and posteromedial walls of the ascending aortic vascular prosthesis, consistent with PVGI. Empirical antibiotic therapy was administered (daptomycin and piperacillin/tazobactam for approximately two weeks, followed by an outpatient course of daptomycin and oral amoxicillin/clavulanic acid for the following 8 months), leading to improvement in inflammatory markers. The follow-up PET/CT scan (8 months later) was similar to the previous one, showing the persistence of focally increased FDG uptake (SUVmax 5.6) despite prolonged antibiotic therapy and normalization of the inflammatory markers (CRP < 0.6 mg/dL versus 2.32 mg/dL, normal value < 0.5; ESR 22 mm/h versus 79 mm/h, normal value < 30). Therefore, WBC scintigraphy was recommended. Planar and SPECT/CT imaging, performed 10 days later, did not reveal any areas of abnormal leukocyte accumulation in the ascending aortic prosthetic walls. WBC scintigraphy was negative for PVGI (Figure 1). The final diagnosis was negative for PVGI, as also confirmed by the absence of symptoms and negative laboratory tests after a one-year follow-up.

Patient no. 2 (72 yrs.) with abdominal aorta prosthesis had fever and decompensated ascites due to toxic cirrhosis, blood culture was positive for *E. coli*, and the CT scan showed a saccular aneurysm of the aortic carrefour; therefore, the patient was referred for [^18^F]FDG PET/CT to rule out the presence of infected aortic aneurysm. [^18^F]FDG PET/CT showed pathologic uptake of [^18^F]FDG (SUVmax 12.5) in a thickened perivascular tissue located cranially to the aortic carrefour, consistent with PVGI. The patient underwent in situ replacement with cryopreserved aortic allografts. Antibiotic therapy with piperacillin/tazobactam was administered for 30 days, then discontinued due to neutropenia, and switched to oral amoxicillin. Daptomycin was added for 8 days due to the positive surgical microbiological culture for *S. epidermidis*. The subsequent follow-up [^18^F]FDG PET/CT scans performed 2 and 3 months later during antibiotic therapy showed the persistence of high [^18^F]FDG uptake in the same perivascular region (SUVmax 6.3 and 5, respectively) despite the good clinical response and the progressive reduction in the inflammatory markers (CRP 0.75 mg/dL versus 1.58 mg/dL, normal value < 0.5). WBC scintigraphy was then recommended. The WBC scan obtained 10 days later (still during amoxicillin therapy) showed minimal leukocyte accumulation, mildly increasing (less than 20%) over time around the posterior periprosthetic region corresponding to the pre-lumbar vertebral region (L4). The WBC scan showed no signs of abnormal leukocyte accumulation around the abdominal aortic periprosthetic tissue, in particular in the thickened perivascular tissue located cranially to the aortic carrefour. This WBC finding was not suggestive of infection but consistent with sterile inflammation. The follow-up WBC scintigraphy, performed 3 months later still during antibiotic therapy (amoxicillin), showed complete disappearance of the prior minimal leukocyte accumulation; thus, being reported as negative for the persistence of PVGI. The final diagnosis at the time of both follow-up nuclear medicine procedures was negative for PVGI, as also confirmed by the absence of symptoms and negative laboratory tests until the end of follow-up.

Patient no. 3 (57 yrs.) had suspected Dacron PVGI of the abdominal aorta. The patient underwent both CTA and [^18^F]FDG PET/CT. CTA showed some signs of infection (fluid collection, gas, and soft tissue enhancement in the periprosthetic region). [^18^F]FDG PET/CT showed abnormally increased uptake of [^18^F]FDG (SUVmax 18.7) along the aorta/right iliac prosthesis and in the periprosthetic collection detected with CTA. CT-guided needle aspiration and microbiological culture were negative for the presence of pathogenic microorganisms. Empirical antibiotic therapy (piperacillin/tazobactam) was administered, and WBC scintigraphy was performed for better characterization of the patient. WBC scintigraphy, performed 5 days later, showed leukocyte accumulation, increasing over time, at the interface between the prosthesis and the native vessel and in the intra-prosthetic region (Figure 2). The WBC findings showed a more limited extension of the infection compared to the PET/CT findings. The patient underwent surgical removal of the infected aortic prosthesis and in situ replacement with cryopreserved aortic allograft and reconstruction with cryopreserved aorto-bisiliac allograft. Microbiological analysis results were positive for *S. warneri* and *S. epidermidis*. Antibiotic therapy (daptomycin, fosfomycin, doxycycline, and long-acting dalbavancin) was prolonged for approximately 4 months until progressive reduction in CRP values and persistent negative blood cultures. Follow-up WBC scintigraphy was performed one month after the end of antibiotic treatment. No abnormal accumulation of leukocytes in periprosthetic regions was detected, scintigraphy thus being consistent with the complete resolution of the infectious disease. The final diagnosis was positive for PVGI with a subsequent complete response to antibiotic treatment.

Patient no. 4 (63 yrs.) had suspected PVGI (endoprosthesis of the descending aorta—Zenith^®^ TX2^®^ Dissection Endovascular Graft with Pro-Form^®^ ref. ZDEG-PT-36-159-PF, Cook Medical, Limerick, Ireland) for FUO, severe neck pain, and positive blood cultures for methicillin-sensitive *Staphilococcus aureus* (MSSA). The patient underwent a complete workup for infective endocarditis. Empirical antibiotic therapy (cefazolin) was then administered. [^18^F]FDG PET/CT, performed to identify the origin of fever, showed markedly increased [^18^F]FDG uptake close to the lower wall of the aortic arch and at the aortic valvular plane. These findings were consistent with possible infection. Two days later, a TEE was performed, proving negative for infective endocarditis but showing a fixed hyperechoic flap in the region of the ascending aortic prosthetic wall. CTA, performed on the same day, showed an abscessual collection adhering to the walls of the descending thoracic aorta, involving the retro-esophageal space, with extrinsic compression in the anterior mediastinum and hypodense formation in the right atrium, compatible with endocarditis vegetation. Antibiotic therapy was implemented with the addition of daptomycin. WBC scintigraphy was performed to confirm and evaluate the extension of infection. The scintigraphy showed, 7 days after CTA, a small but increasing abnormal leukocyte accumulation anterior to the anterior wall of the ascending aortic vascular prosthesis and in the space between the esophagus and the descending aortic prosthesis, without any involvement of the vascular prosthetic walls (Figure 3). Based on the scintigraphic findings, particularly the absence of aortic wall infection, conservative treatment was adopted, continuing antibiotic therapy (dalbavancin and doxycycline) for 20 days. One-month follow-up WBC scintigraphy showed a marked reduction in the intensity and extent of the abnormal leukocyte accumulations, indicating a good response to the antibiotic therapy. The final diagnosis was positive for PVGI with a partial response to antibiotic treatment.

Patient no. 5 (59 yrs.) had a biological aortic valve prosthesis (Edwards Inspiris Resilia 25, Irvine, CA, USA), ascending aorta vascular prosthesis implant (32 mm, Vascutek Ltd., Newmains Avenue, Inchinnan, Renfrewshire PA4 9RR, UK), and chronic post-traumatic left tibial and fibular osteomyelitis. For the onset of FUO and pain in the left lower limb associated with elevated inflammatory markers (ESR 46 mm/h, normal value < 30; CRP 19.8 mg/dL, normal value < 0.5), the patient underwent an [^18^F]FDG PET/CT scan. [^18^F]FDG PET/CT showed increased tracer uptake at the distal medullary and cortical regions of the left tibial diaphysis (SUVmax 7.6) and in the diaphysis of the ipsilateral fibula (SUVmax 4.0). The PET/CT images also showed increased [^18^F]FDG uptake at the right anterolateral wall of the ascending aorta prosthesis (SUVmax 6.2). Three blood cultures obtained in the three subsequent days were negative. WBC scintigraphy was then immediately performed to rule out the presence of ascending aorta PVGI and to distinguish inflammation from infection at the left tibia and fibula. WBC scintigraphy showed leukocyte accumulation, slightly increasing over time (<20%), affecting the distal third diaphysis of the left tibia and the diaphysis of the left fibula, consistent with aseptic chronic osteomyelitis. No leukocyte accumulation was observed at the periprosthetic ascending aorta vascular wall and at the prosthetic aortic valve (Figure 4). The final diagnosis was negative for PVGI, confirmed by the absence of symptoms and negative laboratory tests during the follow-up.

Patient no. 6 (69 yrs.) had a polytetrafluoroethylene (PTFE) femoro-popliteal bypass. One year later, due to the recurrence of critical ischemia of the lower right limb (grade III, category 5 according to the Rutherford scale [16]), a femoral endarterectomy with bovine patch implant and femoro-popliteal subgenicular bypass in the autologous great saphenous vein were performed. The microbiological culture of the proximal region of the PTFE was positive for *S. capitis*. Antibiotic therapy (sulfamethoxazole/trimethoprim) was administered, even though the vascular surgeon and the infectious diseases consultant interpreted the microbiological findings as a false-positive result due to micro-organism contamination during sample collection. For the low probability of graft infection, an [^18^F]FDG PET/CT scan was immediately performed, which showed increased [^18^F]FDG uptake (SUVmax 6.2) along the proximal region of the femoro-popliteal bypass, thus, resulting in suspected infection. The WBC scan was performed 7 days later to rule out or confirm infection. WBC scintigraphy showed leukocyte accumulation, slightly increasing over time (<20%), in the proximal region of the femoro-popliteal bypass. The WBC scan was therefore reported as negative for infection and consistent with inflammation. Antibiotic treatment was discontinued, and the final diagnosis was negative for PVGI.

Patient no. 7 (77 yrs.), with right endovascular treatment using the Viabhan^®^ peripheral endograft (W.L. Gore and Associates Inc., Flagstaff, AZ, USA) and left femoro-popliteal bypass (PTFE). One year later, the patient underwent surgical removal of the right endograft due to infection (MSSA) and limb revascularization with superficial femoro-popliteal bypass in the autologous great saphenous vein. Two months later, the left bypass PTFE was surgically removed, due to infection (*S. aureus* OXAs), combined with left limb revascularization using a femoro-tibio-peroneal bypass in the autologous great saphenous vein. Antibiotic therapy (daptomycin) was administered for 5 months, then daptomycin was switched to sulfamethoxazole/trimethoprim for intolerance. Despite a good clinical and hematological response to antibiotic treatment (CRP negative), the patient suffered from persistent left lower limb edema. The color Doppler echocardiography of the left limb was negative for infection. [^18^F]FDG PET/CT was performed to rule out bypass infection. [^18^F]FDG PET/CT showed focally increased [^18^F]FDG uptake (SUVmax 5.1) along the medial region of the popliteal bypass wall, thus being suspected of infection. The WBC scan was performed 10 days later to distinguish an infectious from a nonspecific cause of [^18^F]FDG uptake. WBC scintigraphy showed mild leukocyte accumulation, stable over time, at the medial region of the popliteal bypass wall; thus, being reported as negative for infection and consistent with inflammation. Antibiotic treatment was discontinued. The final diagnosis was negative for infection.

Patient no. 8 (71 yrs.) had an aorto-bifemoral bypass graft. Four years later, bilateral femoral bifurcation reconstruction using homograft and repair of the left femoral pseudoaneurysm were performed. Another year later, left inguinal collection appeared, with no evidence of pathogens on microbiological culture. CTA was positive for the presence of a pseudoaneurysm at the level of the anastomosis between the left branch of the aorto-bifemoral bypass and the homograft. Empirical antibiotic therapy (sulfamethoxazole/trimethoprim and teicoplanin) was immediately started. The patient underwent a flattening left iliac–femoral pseudoaneurysm with a left common iliac–femoral bypass (Dacron prosthesis), and 10 days later, exclusion of the right iliac anastomotic pseudoaneurysm was performed by placing a covered stent. Eighteen months later, the patient underwent a further procedure for exclusion of the right iliac pseudoaneurysm by placing a covered stent (Viabahn^®^, W.L. Gore and Associates Inc., Flagstaff, AZ, USA). Over the following 2 years, follow-up was negative for recurrent disease, with negative inflammatory markers and CTA showing persistent unchanged collection close to the stent of the left common femoral artery. The last follow-up CTA showed the enlargement of the collection close to the left common femoral artery stent, suspected of infection. The patient was admitted to the Infectious Diseases Department, and antibiotic therapy (cotrimoxazole) was immediately administered. The first microbiological culture of the collection during antibiotic therapy was negative. Antibiotic therapy was discontinued 3 months later due to intolerance. Another microbiological culture of the collection performed 20 days later showed the presence of MSSE. [^18^F]FDG PET/CT performed during antibiotic therapy (teicoplanin) showed focally increased [^18^F]FDG uptake (SUVmax 5.6) around the left common iliac prosthesis, with no abnormal uptake along the entire course of the bilateral aorto-bifemoral vascular prosthesis, particularly at the left femoral periprosthetic collection. The WBC scan (performed 15 days later to distinguish specific from nonspecific [^18^F]FDG uptake) showed leukocyte accumulation, increasing over time, in the left common iliac prosthesis. Additionally, a focal mild leukocyte accumulation, slightly increasing over time, inside the left peri-femoral collection was found. The WBC scan was therefore reported as positive for two foci of infection (Figure 5). The final diagnosis was positive for PVGI (*S. warneri* and *S. epidermidis*). All baseline and follow-up [^18^F]FDG PET/CT and WBC scintigraphy were classified as positive or negative according to the described interpretation criteria. No equivocal results were found. Overall, we evaluated 11 [^18^F]FDG PET/CT and 11 WBC scintigraphies. All the baseline [^18^F]FDG PET/CT values were positive: three true-positive (TP) and five false-positive (FP). Three follow-up [^18^F]FDG PET/CT scans were performed in two patients (two in one patient and one in the other patient), resulting in three false-positive scans. The baseline WBC scintigraphies were positive in three patients (TP) and negative in five patients (true negative: TN); the three follow-up WBC scintigraphies performed in three patients were positive in one patient (TP) and negative in the other two patients (TN) (Figure 6).

## 4. Discussion

Major and minor MAGIC criteria are the currently employed diagnostic criteria utilized for PVGI. They are based on clinical/surgical, radiological, and microbiology/laboratory findings [8]. Based on one major criterion or two minor criteria, the presence of infection is suspected. Apart from direct intraoperative identification of pus around a graft and positive micro-organism culture, the MAGIC criteria predominantly rely on indirect evidence of aortic graft infection, such as the presence of gas or fluid around a graft on CTA [17]. The sensitivity and specificity of CTA for diagnosing PVGI infection are high in patients with a high pre-test probability of infection [2,6]. In low-grade graft infections, however, the sensitivity drops while specificity remains high [2,6,18,19]. Nuclear medicine procedures can improve the diagnostic accuracy of radiologic imaging to avoid therapy failures in patients with PVGI. [^18^F]FDG PET/CT is a valuable tool for diagnosing PVGI. The [^18^F]FDG uptake pattern seems to be the most accurate assessment method for diagnosing PVGI, a focal, non-homogeneous pattern of increased [^18^F]FDG uptake being considered a reliable biomarker of graft infection [20,21]. The combination of major CT criteria and focal tracer uptake (versus diffuse uptake) can help diagnose infection, thus improving the overall diagnostic accuracy [13]. [^18^F]FDG PET/CT interpretation requires caution, however; in particular, the main and non-negligible pitfall of [^18^F]FDG PET/CT lies in the impossibility of differentiating between infection and inflammation, as there is no definite association between infection and increased uptake value (expressed as SUVmax) of [^18^F]FDG [6,20]. It is therefore recommended to postpone the use of [^18^F]FDG PET/CT imaging to at least 8–12 weeks after implantation to reduce false-positive findings related to the healing process and inflammation [18,22]. The type of procedure performed (endovascular implantation vs. open surgical repair) should also be considered. A threshold value of [^18^F]FDG SUVmax equal to 4.5 was proposed for endovascular repairs, while ‘stepwise threshold values’ were proposed based on the time elapsed since surgery in cases of open surgery (SUVmax 6.5 up to 1 year, 5.5 up to 3 years, and 5.0 after 5 years) [23]. In our patients (cases 1, 2, 5, 6, 7), with positive [^18^F]FDG PET/CT (FP) and negative WBC scintigraphy (TN), the SUVmax values were higher than the threshold of 4.5 (5.2, 12.5, 6.2, 6.2, and 5.1, respectively). Although based on a small cohort of patients, our data suggest that even in cases of endovascular prosthetic implantation, it is not appropriate to rely solely on a threshold value of SUVmax for distinguishing between infection and inflammation of the vascular graft. WBC scintigraphy has high diagnostic accuracy, especially if including SPECT/CT imaging [24], and it can be employed at any time after surgery [5]. Planar images of WBC scintigraphy are acquired after 30 min, 3–4 h, and 18–20 h after injection. SPECT acquisitions, 3D reconstruction, and fusion with low-dose CT should be acquired after delayed and late planar images. Late SPECT/CT images are crucial, even if the imaging quality can be lower than at the delayed SPECT/CT acquisition, by enabling one to assess the pattern of leukocyte accumulation over time. SPECT/CT acquisition is, moreover, mandatory, since a pathologic accumulation of WBC in thoracic or abdominal vascular prosthesis can be masked on planar images by physiologic accumulation in the sternum, vertebrae, or bowel. WBC scintigraphy has higher specificity than [^18^F]FDG PET/CT in identifying active infections, especially with the use of SPECT/CT acquisitions that lead to a significant reduction in the number of false-positive and false-negative results [9,25,26]. The specificity of this test depends on the progressive accumulation of neutrophil granulocytes, which is higher in acute infection, lower in subacute disease, and mild and stable in chronic inflammation. A systematic review and meta-analysis showed a pooled 0.67 sensitivity (95% CI 0.57–0.75) of CTA versus 0.95 (95% CI 0.87–0.99) for [^18^F]FDG PET/CT and 0.99 (95% CI 0.92–1.00) for WBC scintigraphy with SPECT/CT. The pooled specificity was 0.63 (95% CI 0.48–0.76) for CTA, 0.70 (95% CI 0.59–0.81) for [^18^F]FDG PET/CT, and 0.82 (95% CI 0.57–0.96) for WBC scintigraphy with SPECT/CT [3,6,7]. Similar values of sensitivity, specificity, and accuracy of contrast-enhanced CT and [^18^F]FDG PET/CT have been confirmed by Husmann et al. [27]. Our cases showed concordant results in three patients and discordant results in five patients. Among the patients with concordant results, WBC scintigraphy showed a more precise location of infection compared to [^18^F]FDG PET/CT. In two of these patients, the extension of [^18^F]FDG uptake has been wider and somewhat nonspecific compared to pathologic leukocyte accumulation. In this regard, we would underline the importance of a multidisciplinary approach during the diagnostic work-up and [^18^F]FDG PET/CT interpretation. For example, in patient no. 4, according to the [^18^F]FDG PET/CT findings, the surgical procedure should be performed; however, the WBC scintigraphy findings allowed for the maintenance of the conservative approach of continuing antibiotic therapy. In the other patient with a highly suspected infection of the left common iliac–femoral bypass, we found concordant results for both [^18^F]FDG PET/CT and WBC scintigraphy in one site of the bypass (left common iliac tract), while [^18^F]FDG PET/CT was negative (FN) and the WBC scan was positive (TP) for infection in another site of the bypass (proximal femoral collection) due to the poor vascularization of the collection. All five patients with discordant results had a negative final diagnosis for PVGI. In these patients, [^18^F]FDG PET/CT showed high [^18^F]FDG uptake with SUVmax higher than 4.5, whereas all WBC scintigraphies were negative. This small clinical case collection confirms that nuclear imaging techniques such as [^18^F]FDG PET/CT and WBC scintigraphy can improve the diagnostic accuracy of CTA in PVGI, especially in mild and low-grade infections, which are often overlooked or misdiagnosed by CTA. According to MAGIC criteria [8], five patients (cases 1, 2, 5, 6, and 7) can be classified as ‘suspected’ PVGI and three patients as ‘diagnosed’ PVGI (cases 3, 4, and 8). It is important to point out that only in the patients with ‘diagnosed infection’ did both PET/CT and WBC scintigraphy yield concordant results (three TP PET/CT and three TP WBC scintigraphies), while in patients with ‘suspected infection’, we found eight false-positive PET/CT (five baselines and three follow-ups) and six true-negative WBC scintigraphies (five baselines and one follow-up). Therefore, we can state that, although [^18^F]FDG-PET/CT has a very high sensitivity and moderate specificity for detecting PVGI [28,29], WBC scintigraphy shows greater specificity, proving to be essential in the management of patients with vascular prosthesis infection, especially in the case of persistence of no specific high uptake of [^18^F]FDG despite the good clinical response to therapy. However, it should not be forgotten that WBC scintigraphy is more time-consuming than PET/CT, and some patients cannot be cooperative enough to perform it. Furthermore, a WBC scan requires specialized personnel, blood manipulation, and suitable equipment for leukocyte labeling.

## 5. Conclusions

This small case series of patients contributes to the literature by confirming the better diagnostic accuracy of WBC scintigraphy compared to [^18^F]FDG PET/CT, especially in all infected patients who need an evaluation of their treatment response. We suggest to perform WBC scintigraphy, integrating planar with SPECT/CT images, to allow a better anatomical location of findings. It therefore provides clinicians/surgeons with a valid tool for choosing the most appropriate treatment in patients with VPJI. The diagnostic accuracy of [^18^F]FDG PET/CT significantly decreases during the follow-up of infected patients. The remarkably high diagnostic accuracy of WBC scintigraphy, if confirmed in further studies with a large number of patients, could be used for the evaluation of antibiotic therapy response, and WBC scintigraphy might change its position from minor to major MAGIC criteria. We think that it is good to trust [^18^F]FDG PET/CT in PVGI diagnosis, but to check is better.

## Figures and Tables

**Figure 1 jcm-14-04352-f001:**
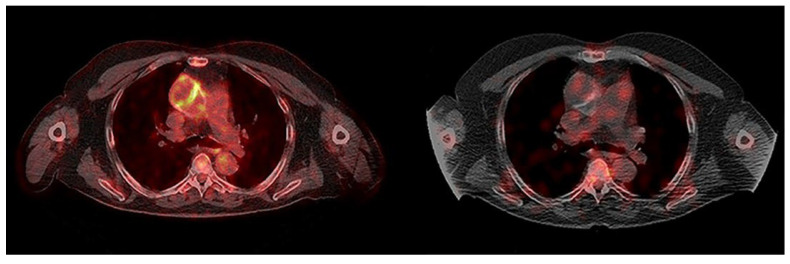
**Discordant findings in a patient with no PVGI.** (**Left**) The transaxial fused PET/CT image shows high FDG uptake at the posteromedial wall of the ascending aorta and small focal uptake on its anterior wall; (**right**) the transaxial fused late SPECT/CT image of WBC scintigraphy shows no pathological WBC uptake at the ascending aorta’s wall.

**Figure 2 jcm-14-04352-f002:**
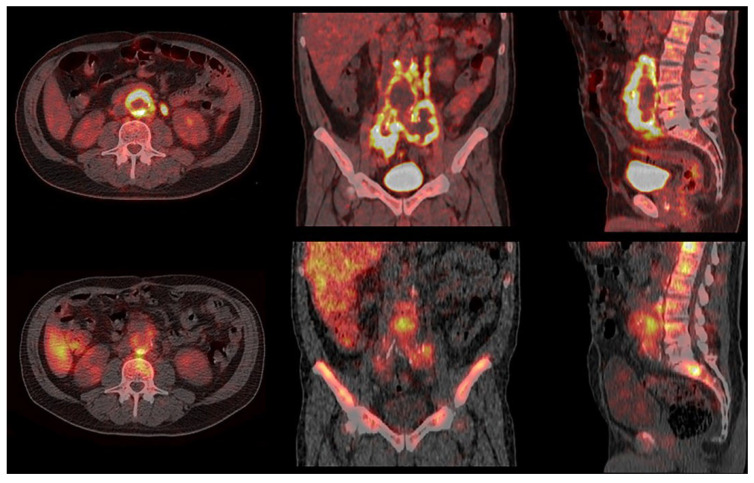
**Discordant findings in a patient with PVGI.** Top: Transaxial (**left**), coronal (**middle**), and sagittal (**right**) fused PET/CT images show intense uptake of FDG at the periprosthetic vascular abdominal aortic anterior wall, extending to the iliac vessels and both ileo-psoas muscles; bottom: transaxial (**left**), coronal (**middle**), and sagittal (**right**) fused late SPECT/CT images of WBC scintigraphy show pathological leukocyte accumulation only to the native vessel/prosthesis posterior interface and to the corresponding intra-prosthetic abdominal aortic site with minimal involvement of the left common iliac vessel and both ileo-psoas muscles.

**Figure 3 jcm-14-04352-f003:**
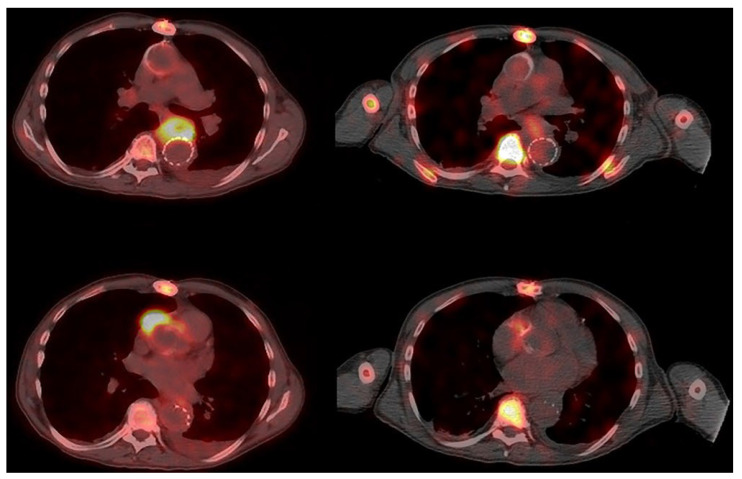
**Discordant findings in a patient with PVGI.** Left: Transaxial fused PET/CT images show diffuse high uptake of FDG at the periprosthetic vascular anterior wall of the descending aorta (**top**) and intense uptake of FDG in front of the anterior wall of the vascular ascending aortic prosthesis (**bottom**); right: transaxial fused late SPECT/CT images of WBC scintigraphy show a mild pathologic accumulation of WBC in front of the anterior wall of the vascular descending aortic prosthesis (**top**) and in front of the anterior wall of the vascular ascending aortic prosthesis (**bottom**).

**Figure 4 jcm-14-04352-f004:**
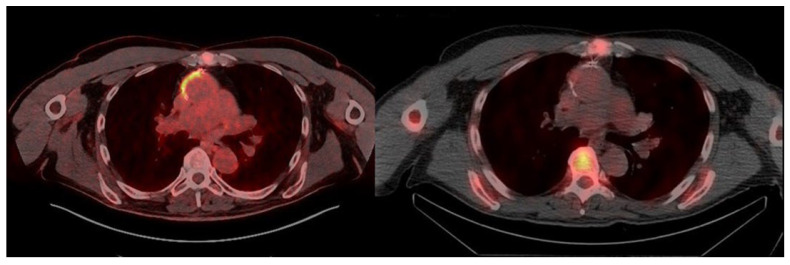
**Discordant findings in a patient with no PVGI.** (**Left**) The transaxial fused PET/CT image shows moderate FDG uptake at the anterolateral wall of the ascending aorta; (**right**) the transaxial fused late SPECT/CT image of WBC scintigraphy shows no pathologic WBC uptake at the ascending aorta’s wall.

**Figure 5 jcm-14-04352-f005:**
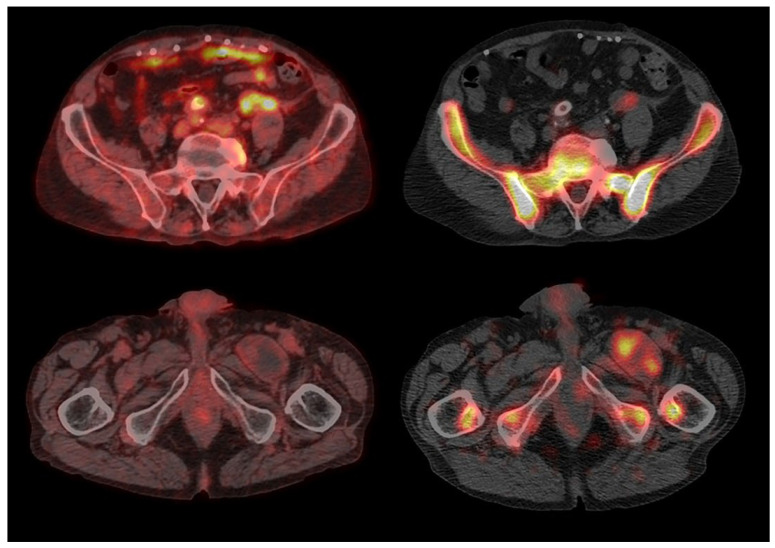
**Discordant findings in a patient with PVGI.** (**Left**) transaxial fused PET/CT images show diffuse high uptake of FDG around the left common iliac prosthesis (**top**) with no pathological uptake along the entire course of the bilateral aorto-bifemoral vascular prosthesis, particularly at the left femoral periprosthetic collection (**bottom**); right: transaxial fused late SPECT/CT images of WBC scintigraphy show an increased leukocyte accumulation in the left common iliac prosthesis (**top**) and a mild pathologic accumulation into the left femoral periprosthetic collection (**bottom**).

**Figure 6 jcm-14-04352-f006:**
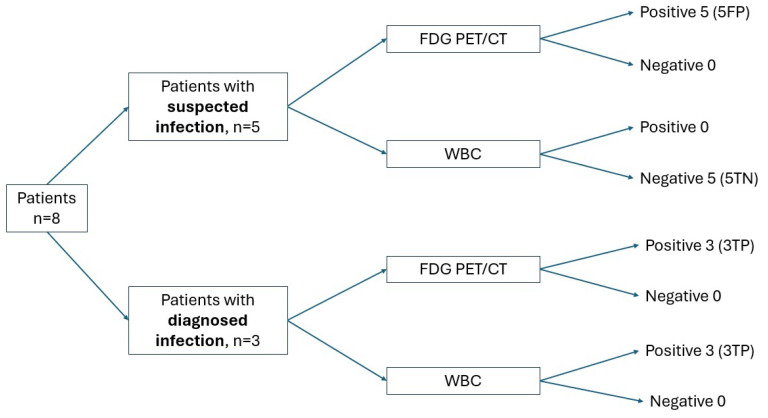
Diagnostic performance of [^18^F]FDG PET/CT and WBC scintigraphy in patients with suspected and diagnosed infections according to MAGIC criteria.

**Table 1 jcm-14-04352-t001:** Clinical characteristics, graft location, identified microorganisms, CTA results, and final diagnosis in eight patients with suspected vascular graft infection. CTA = computed tomography angiography; c.m. = contrast medium; MSSA = methicillin-sensitive staphylococcus aureus; OXAs = oxacillin-sensitive; “+” indicates a positive CTA suggestive of infection.

Patient No.	Age	Comorbidities	Site of the Graft	Microorganisms	CTA	Final Diagnosis
1	69	//	Thoracic aorta	//	Not performed (allergy to c.m.)	Not infected
2	72	//	Abdominal aorta	*E. coli*	Not diagnostic	Not infected
3	57	//	Aorto-bisiliac	*S. warneri* and *S. epidermidis*	+	Infected
4	63	Marfan syndrome	Thoracic aorta	MSSA	+	Infected
5	59	//	Thoracic aorta	//	Not performed	Not infected
6	69	//	Lower limb	*S. capitis*	Not performed (renal failure)	Not infected
7	77	//	Lower limb	MSSA and *S. aureus* OXAs	Not performed (allergy to c.m.)	Not infected
8	71	Diabetes mellitus	Aorto-left femoral	*S. epidermidis* erythromycin/clindamycin resistant	+	Infected

## Data Availability

The raw data supporting the discussion in this article will be made available by the authors upon request.

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
