# Peer review of "A Head-to-Head Comparison Between [18F]Fluorodeoxyglucose ([18F]FDG) Positron Emission Tomography/Computed Tomography (PET/CT) and 99mTechnetium-Hexamethylpropylene Amine Oxime (HMPAO)-Labeled Leukocyte Scintigraphy in a Case Series of Patients with Suspected Vascular Prosthesis Infection: To Trust Is Good, but to Check Is Better"

_jcm, 2025, doi:10.3390/jcm14124352_

Round 1
Reviewer 1 Report
Comments and Suggestions for Authors
The manuscript reports the causes of morbidity and mortality in 8 patients with vascular prostheses.
- Manuscript title, try to change “[18F]FDG PET/CT and 99mTc-2 HMPAO” for full names, because not everybody knows these acronymous
- Abstract: Background., prepare abstract according to author instructions, rewrite abstract only with the important experimental and important results
- Line 92 “We retrospectively studied 8 adult patients” including information if who number are male or female. Include information from line 160 in this part
- Why was the manuscript prepared only with 8 patients information, was representative?
- Lines 102-105 “Exclusion criteria” improve this part
- Line 191 “E. coli” correct write is with cursive or italics
- Lines 351-352 “All patients were classified as positive or negative according to the described interpretation criteria.” Who is classified?
- Caption of Table 1 must be top of the Table
- Figure 6, why selected Patient 8
- 5. Conclusions, needs to improve
Author Response
Comment 1: Manuscript title, try to change “[18F]FDG PET/CT and 99mTc-2 HMPAO” for full names, because not everybody knows these acronymous.
Response 1: Thank you for pointing this out. As suggested, the title has been modified accordingly, as reflected in lines 1 to 7.
Comment 2: Abstract: Background., prepare abstract according to author instructions, rewrite abstract only with the important experimental and important results.
Response 2: Thank you for your helpful suggestion. The abstract has been revised accordingly and can be found in lines 17–31.
Comment 3: Line 92 “We retrospectively studied 8 adult patients” including information if who number are male or female. Include information from line 160 in this part.
Response 3: We have included the patient information as suggested, in lines 84–85.
Comment 4: Why was the manuscript prepared only with 8 patients information, was representative?
Response 4: Thanks for this valuable comment. Our study included only 8 of more than 20 patients studied in 2022 July-2023 December because the described restrictive inclusion and exclusion criteria can ensure the data homogeneity and comparability. Even if the small number of patients may not be representative of the larger population, this study allows to understand the advantages and limits of both nuclear medicine techniques in PVGJ diagnosis. We reviewed and emphasized in the 'discussion' and 'conclusion' sessions that further studies with larger cohorts of patients are mandatory to confirm our findings.
Comment 5: Lines 102-105 “Exclusion criteria” improve this part.
Response 5: Thank you for your suggestion. We have attempted to improve the "Exclusion Criteria" paragraph as recommended, and the revised version can be found in lines 93–100.
Comment 6: Line 191 “E. coli” correct write is with cursive or italics
Response 6: Thank you. We have made the adjustment as suggested (see line 186).
Comment 7: Lines 351-352 “All patients were classified as positive or negative according to the described interpretation criteria.” Who is classified?
Response 7: Thank you, we have clarified the sentence as suggested (lines 346–347).
Comment 8: Caption of Table 1 must be top of the Table.
Response 8: Thank you, we have placed the table caption above the table as suggested (lines 357–360).
Comment 9: Figure 6, why selected Patient 8.
Response 9: Figure 6 summarizes more directly the results of the study, it can be removed if you suggest.
Comment 10: Conclusions, needs to improve
Response 10: Thank you for the suggestion. We have improved the conclusions accordingly (lines 489–499).
Reviewer 2 Report
Comments and Suggestions for Authors
In this work titled -Head-to-head comparison between [18F]FDG PET/CT and 99mTc- 2 HMPAO labeled leukocyte imaging in a case series of patients 3 with suspected vascular prosthesis infection: to trust is good 4 but to check it out is better-the authors showed retrospectiv evaluation of 8 patients with suspected s. Prosthetic vascular graft infection (PVGI) who underwent 24 both [18F]FDG PET/CT and white blood cells scintigraphyWBC scintigraphy.
The research is interesting. The work is done well and is well presented. All results and experimental setup is proper. However references are too old, please refresh.
Also conclusion too short and not relevant, please edit.
In this work titled -Head-to-head comparison between [18F]FDG PET/CT and 99mTc- 2 HMPAO labeled leukocyte imaging in a case series of patients 3 with suspected vascular prosthesis infection: to trust is good 4 but to check it out is better-the authors showed retrospectiv evaluation of 8 patients with suspected s. Prosthetic vascular graft infection (PVGI) who underwent 24 both [18F]FDG PET/CT and white blood cells scintigraphyWBC scintigraphy.
The research is interesting. The work is done well and is well presented. All results and experimental setup is proper. However references are too old, please refresh.
Also conclusion too short and not relevant, please edit.
Thank you
Author Response
Comment 1: references are too old, please refresh.
Response 1: Thank you for this important suggestion. We have updated the bibliography accordingly (lines 519-588).
Comment 2: Also conclusion too short and not relevant, please edit.
Response 2: Thank you for your suggestion. We have worked to enhance the conclusions accordingly (lines 489-499).
Round 2
Reviewer 2 Report
Comments and Suggestions for Authors
Thank you